# Plasma Metabolomics Predicts Chemotherapy Response in Advanced Pancreatic Cancer

**DOI:** 10.3390/cancers15113020

**Published:** 2023-06-01

**Authors:** Hayato Muranaka, Andrew Hendifar, Arsen Osipov, Natalie Moshayedi, Veronica Placencio-Hickok, Nicholas Tatonetti, Aleksandr Stotland, Sarah Parker, Jennifer Van Eyk, Stephen J. Pandol, Neil A. Bhowmick, Jun Gong

**Affiliations:** 1Department of Medicine, Cedars-Sinai Medical Center, Los Angeles, CA 90048, USA; hayato.muranaka@cshs.org (H.M.); andrew.hendifar@cshs.org (A.H.); arsen.osipov@cshs.org (A.O.); natalie.moshayedi@cshs.org (N.M.); veronica.placencio-hickok@cshs.org (V.P.-H.); stephen.pandol@cshs.org (S.J.P.); neil.bhowmick@cshs.org (N.A.B.); 2Samuel Oschin Comprehensive Cancer Institute, Cedars-Sinai Medical Center, Los Angeles, CA 90048, USA; 3Department of Computational Biomedicine, Cedars-Sinai Medical Center, Los Angeles, CA 90048, USA; nicholas.tatonetti@cshs.org; 4Advanced Clinical Biosystems Research Institute, Cedars-Sinai Medical Center, Los Angeles, CA 90048, USA; aleksandr.stotland@cshs.org (A.S.); sarah.parker@cshs.org (S.P.); jennifer.vaneyk@cshs.org (J.V.E.); 5Department of Biomedical Sciences, Cedars-Sinai Medical Center, Los Angeles, CA 90048, USA; 6Department of Research, VA Greater Los Angeles Healthcare System, Los Angeles, CA 90073, USA

**Keywords:** pancreatic cancer, chemotherapy, FOLFIRINOX, gemcitabine/nab-paclitaxel, plasma metabolomics

## Abstract

**Simple Summary:**

Pancreatic cancer is a clinically heterogeneous disease, and treatment leads to different outcomes for people with similar diagnoses. Identifying patients who may respond to chemotherapy and thereby benefit from improved survival has important implications for treatment protocols. In this study, plasma metabolite profiling was performed to identify potential biomarker candidates that can predict the response of patients to different neoadjuvant chemotherapy regimens for pancreatic cancer. The concentrations of several metabolites from LC–MS were significantly different when comparing the response to chemotherapy. Several metabolites demonstrated a predictive performance for the response to chemotherapy. These results show promise for larger studies that would validate the findings, which could contribute to the development of more personalized treatment protocols for pancreatic cancer patients.

**Abstract:**

Pancreatic cancer (PC) is one of the deadliest cancers. Developing biomarkers for chemotherapeutic response prediction is crucial for improving the dismal prognosis of advanced-PC patients (pts). To evaluate the potential of plasma metabolites as predictors of the response to chemotherapy for PC patients, we analyzed plasma metabolites using high-performance liquid chromatography–mass spectrometry from 31 cachectic, advanced-PC subjects enrolled into the PANCAX-1 (NCT02400398) prospective trial to receive a jejunal tube peptide-based diet for 12 weeks and who were planned for palliative chemotherapy. Overall, there were statistically significant differences in the levels of intermediates of multiple metabolic pathways in pts with a partial response (PR)/stable disease (SD) vs. progressive disease (PD) to chemotherapy. When stratified by the chemotherapy regimen, PD after 5-fluorouracil-based chemotherapy (e.g., FOLFIRINOX) was associated with decreased levels of amino acids (AAs). For gemcitabine-based chemotherapy (e.g., gemcitabine/nab-paclitaxel), PD was associated with increased levels of intermediates of glycolysis, the TCA cycle, nucleoside synthesis, and bile acid metabolism. These results demonstrate the feasibility of plasma metabolomics in a prospective cohort of advanced-PC patients for assessing the effect of enteral feeding as their primary source of nutrition. Metabolic signatures unique to FOLFIRINOX or gemcitabine/nab-paclitaxel may be predictive of a patient’s response and warrant further study.

## 1. Introduction

Pancreatic cancer (PC) is one of the deadliest cancers, with a rising incidence over the past two decades [1]. The combined majority (nearly 80%) of patients diagnosed with PC will have unresectable, locally advanced PC (LAPC) or metastatic PC (collectively termed advanced PC or APC) [2,3]. The 5-year survival for all stages of PC approximates 11%, and the prognosis is dismal for metastatic PC, where the 5-year survival approximates 3% [3]. The clinical course for PC is often compounded by progressive therapeutic resistance and cachexia, wherein nearly 85% of patients with PC meet the classical definition of cancer cachexia [4]. Systemic therapy combinations such as FOLFIRINOX (fluorouracil, leucovorin, irinotecan, and oxaliplatin) and gemcitabine hydrochloride plus nanoparticle albumin-bound paclitaxel (gemcitabine + abraxane) represent standard therapies that have been shown to prolong the survival of patients with metastatic PC [2]. However, progression to systemic therapies is often the rule, rather than exception, in metastatic PC, with a clinical course that is often marked by therapeutic resistance to subsequent lines of therapy following the failure of first-line chemotherapy [5]. As such, there is a high unmet need to develop biomarkers that can predict a patient’s response to treatment or their prognosis in APC.

Metabolomics enable the quantitative detection of multiple small-molecule metabolites in cells, tissues, and biofluids in combination with advanced bioinformatics approaches [6,7]. The high-throughput analytical techniques of nuclear magnetic resonance (NMR) spectroscopy and mass spectrometry (MS) combined with multivariate statistical analyses provide information on a large number of metabolites, including those that have altered levels between healthy subjects and patients with various diseases, including cancer [8]. A major advantage in the application of metabolomics comes from an improved ability to detect thousands of metabolites in parallel, which could help us to monitor the dynamic picture of disease progression [9,10]. In the past few years, ultra-performance liquid chromatography coupled to time-of-flight mass spectrometry (LC–MS) has become one of the most advanced and useful tools. So far, metabolomics-based approaches have been used in a large variety of applications, including early disease detection, drug response analyses, toxicity and nutritional studies, and basic systems biology [9,11]. Compared with other biomarker discovery approaches for cancer, metabolomics provide a strong link between the genotype and phenotype, and may provide more insight into oncogenesis. Importantly, once established, tests based on metabolic profiles are relatively inexpensive, are rapid, and can be automated [12].

A growing number of metabolomics studies are contributing toward an improved understanding of PC. Previous studies have examined the metabolites using pancreatic tumor tissues and normal adjacent tissues [13,14,15], urine [16,17,18], serum [19,20,21,22,23,24], and plasma [25,26,27,28], and have identified a metabolic signature in pancreatic cancer. These results demonstrated that the metabolites involved in the metabolism of lipids, glucose, amino acids, choline, DNA synthesis, small organic acids, and muscle protein breakdown can discriminate pancreatic cancer from healthy controls and chronic pancreatitis [29,30,31]. These findings may allow earlier and more precise diagnostics, prognostics, and the prediction of new therapeutic targets, enabling a potential application in personalized therapy for pancreatic cancer. However, these studies have almost entirely focused on the early detection of pancreatic cancer, with only a few studies aimed at biomarker discovery for predicting the response to chemotherapy in pancreatic cancer patients.

In this study, we conducted plasma metabolomics on prospectively collected blood samples from a cohort of patients with APC enrolled into a single-arm clinical trial of enteral feeding and standard chemotherapy. The goal was to determine the feasibility of measuring potential biomarkers that could predict the response to chemotherapy. Our results show promise for future studies that could provide more insight into optimizing chemotherapy for PC patients using metabolic biomarkers.

## 2. Materials and Methods

### 2.1. Study Populations

The PanCax-1 was a single-arm, single-institution prospective trial (NCT02400398) that enrolled 36 pts with APC meeting consensus criteria for cachexia [32,33]. The original study design, eligibility criteria, and primary analysis results have been previously published [32,33]. Briefly, 31 patients with unresectable LAPC or metastatic PC planned for standard-of-care chemotherapy were enrolled and underwent jejunal feeding tube placement to receive enteral feeding (Peptamen 1.5) over 12 weeks (three 28-day cycles). The primary outcome was weight stability at 12 weeks (cycle 3), which was defined as a weight change of less than 0.1 kg/baseline body mass index (BMI) unit. As part of preplanned exploratory analyses, blood samples were collected during the 12-week enteral feeding period, processed to plasma, and stored at −80 degrees Celsius. Plasma metabolomics were conducted under an institutional review board (IRB)-approved protocol (STUDY00000250).

### 2.2. Study Design

This was a retrospective analysis on prospectively collected blood samples from 31 subjects enrolled into the PanCax-1 prospective trial (NCT02400398). As part of study correlatives, blood samples were collected at a baseline or time 0 and after 6 weeks and 12 weeks of enteral feeding (three 28-day cycles) in cachectic patients with APC planned for standard chemotherapy. Patients were stratified by stable disease (SD), partial response (PR), or progressive disease (PD) as the best overall response to a standard chemotherapy regimen. The standard chemotherapy regimens included FOLFIRINOX (fluorouracil, leucovorin, irinotecan, and oxaliplatin), FOLFIRI (fluorouracil, leucovorin, and irinotecan), gemcitabine, and gemcitabine and abraxane. One patient received gemcitabine + abraxane + peg-hyaluronidase under a clinical trial. For focused metabolite analyses, the samples were stratified into pretreatment and FOLFIRINOX-, FOLFIRI-, or gemcitabine + abraxane-treated samples.

### 2.3. Metabolic Analysis (dMRM; Dynamic Multiple-Reaction Monitoring)

Plasma metabolites were extracted by adding 4 volumes of a 50:50 methanol:ethanol extraction buffer. The samples were vortexed, cleared by centrifugation at 21,000× *g* for 10 min at 4 °C, and the metabolite-containing supernatant was dried with a Thermo Fisher (Waltham, MA, USA) Savant SPD1010-115 SpeedVac benchtop centrifugal concentrator. The metabolite extract was resuspended in a 20% methanol and 80% water buffer for the analysis. The extractions were analyzed with an Agilent 6470A triple quadrupole mass spectrometer operating in negative mode, connected to an Agilent 1290 ultra high-performance liquid chromatography (UHPLC) system (Agilent Technologies, Santa Clara, CA, USA). The mobile phases consisted of HPLC- or LCMS-grade reagents. Buffer A was water with 3% methanol, 10 mM tributylamine (TBA), and 15 mM acetic acid. Buffers B and D were isopropanol and acetonitrile, respectively. Finally, Buffer C was methanol with 10 mM TBA and 15 mM acetic acid. The analytical column used was an Agilent ZORBAX RRHD Extend-C18 1.8 µm 2.1 × 150 mm coupled with a ZORBAX Extend Fast Guard column for UHPLC Extend-C18, 1.8 µm, 2.1 mm × 5 mm. The MassHunter Metabolomics dMRM Database and Method was used to scan for up to 219 polar metabolites within each sample (Agilent Technologies, Santa Clara CA, USA). The resulting chromatograms were visualized using Agilent MassHunter Quantitative Analysis for QQQ. The final peaks were manually checked for consistent and proper integration.

Metabolites were extracted from plasma and analyzed with a dedicated triple quadrupole LC–MS (Agilent Technologies, Santa Clara, CA, USA) where up to 219 polar metabolites within each sample were measured by the relative area under the curve (AUC).

### 2.4. Statistical Analysis

The metabolic profiles of plasma from pancreatic cancer patients with a partial response (PR)/stable disease (SD) or progressive disease (PD) to chemotherapy drugs were studied using a combination of LC–MS and multivariate analysis methods. The predictive performance was evaluated in terms of sensitivity, specificity, and accuracy based on the random forest prediction model and the leave-one-out cross-validation. A Student’s *t*-test was performed between responder (SD/PR) and non-responder (PD) samples. A *p*-value < 0.05 was considered significant. A partial least-squares discriminant analysis (PLS-DA) was employed to present the discrimination performance of metabolites between SD/PR and PD samples, and the variable importance in the project (VIP) was calculated as well. Potential metabolic biomarkers were selected with the criteria of VIP > 1 and *p* < 0.05. A random forest (RF) prediction model was constructed with the potential biomarkers. The predictive performance for the potential biomarkers was evaluated by the sensitivity, specificity, accuracy, and area under the receiver-operating characteristic (ROC) based on the leave-one-out cross-validation (CV). Metaboanalyst v5.0 was used to generate a PLS-DA score plot and ROC curves. Descriptive statistics were employed when indicated and expressed as the mean or median and frequencies as percentages. All statistical analyses were performed using GraphPad Prism 7 (GraphPad Software, San Diego, CA, USA).

## 3. Results

### 3.1. Patients and Clinical Characteristics

The original PanCax-1 trial (NCT02400398) consisted of 31 patients with APC between April 2015 and March 2019. In this single-institution, single-arm prospective trial, 31 cachectic patients with APC underwent jejunal tube placement to receive 12 weeks of enteral nutrition and were planned for standard chemotherapy [32,33]. As part of preplanned exploratory analyses, blood samples were prospectively collected across three 28-day cycles of enteral feeding. The median age was 67.1 years with 19/31 (61.3%) females. There were 61 blood samples collected from 31 patients with APC that were available for plasma metabolomics. A total of 22/32 (68.8%) patients received first-line chemotherapy, the majority of whom (18/22 (81.8%)) received gemcitabine-based chemotherapy. A total of 9/33 (27.3%) patients received 5-fluorouracil (5-FU)-based chemotherapy. There were 2/32 with partial responses (PRs, 6.3%) and 10/32 with stable disease (SD, 31.3%) as the best response to chemotherapy. The median overall survival (OS) for the cohort was 6.53 months; for no treatment, it was 4.58 months; for gemcitabine-based chemotherapy, it was 8.83 months; and for 5-FU-based chemotherapy, it was 5.48 months (Table 1). Kaplan–Meier plots of the cohort are shown in Appendix A.

### 3.2. Plasma Metabolomic Profiles from APC Patients with Different Overall Response to Standard Chemotherapy Regimen

In order to visualize the classification performance of the metabolic profiling, a partial least-squares discriminant analysis (PLS-DA) score plot is depicted (Figure 1). Although the principal component analysis (PCA) score plot based on the metabolomics data revealed a poor separation (Appendix A), the PLS-DA showed a clear separation between PR/SD and PD in the pretreatment patients and the patients with FOLFIRINOX or FOLFIRI (Figure 1A,B). The score plot of PLS-DA in the patients who received gemcitabine + abraxane showed a slight overlap between PR/SD and PD (Figure 1C). Furthermore, a heatmap showed the different trends in metabolic fluctuates throughout various chemo treatments and responses to chemotherapy (Appendix A). These results indicate that plasma metabolic profiles might reflect the chemotherapy regimen as well as the overall response to chemo treatments.

### 3.3. The Discovery and Identification of Metabolic Biomarkers

We employed a multivariate analysis approach and Student’s *t*-test to identify the set of metabolites that had the highest independent ability to predict the response to chemotherapy treatment. The potential biomarkers were selected with the criteria of *p* < 0.05 and VIP > 1. The bar graphs for significantly altered metabolites (*p* < 0.05) and the top 20 VIP scores are shown in Figure 2. Additionally, classical volcano plots with log2 (fold-change) on the *X*-axis against -log10 (*p*-value) from the *t*-test on the *Y*-axis are shown in Appendix A. Appendix A display the fold-change, *p*-value, and VIP score for all the metabolites detected in the plasma samples.

In the pretreatment patients, eight biomarkers reached statistical significance and the concentrations of taurocholic acid, inosine 5-triphosphate, inosine, and deoxyguanosine 5-triphosphate were consistently increased at baseline in the subjects who experienced PD as the best response compared to those with SD + PR, while concentrations of l-asparagine, uric acid, phenylpyruvic acid, and l-cystine were decreased in those with PD compared to SD + PR at baseline (Figure 2A).

In patients treated with FOLFIRINOX or FOLFIRI, 13 biomarkers were significantly different between responders (SD + PR) and non-responders (PD). The concentrations of 2-deoxyguanosine 5-monophosphate, 2-phosphoglyceric acid, adenosine 5-diphosphate, l-arginine, l-asparagine, l-leucine, l-methionine, l-serine, l-tryptophan, l-tyrosine, *N*-acetylglutamic acid, *N*-carbamoyl-dl-aspartic acid, and pyridoxal hydrochloride were decreased in those experiencing PD compared to SD + PR as the best response (Figure 2B).

In patients treated with gemcitabine + abraxane, 11 biomarkers were significantly different between responders (SD + PR) and non-responders (PD), wherein the concentrations of 2-methyl-1-butanol, hypoxanthine, lactic acid, l-glutamic acid, pyruvic acid, taurocholic acid, and xanthine were consistently increased in those with PD compared to those with SD + PR as the best response, while concentrations of l-isoleucine, o-hydroxy hippuric acid, and salicylic acid were decreased for those experiencing PD compared to SD + PR as the best response (Figure 2C).

### 3.4. Biomarkers for the Prediction of a Response to Chemotherapy

Next, we evaluated the predictive performance of metabolites individually using the biomarker analysis module of Metaboanalyst v5.0 https://www.metaboanalyst.ca (accessed on 23 May 2023). A random-forest-based multivariate receiver-operating characteristic (ROC) analysis using *t*-statistics for metabolite ranking resulted in the panels illustrated in Figure 3 and Figure 4. In patients treated with FOLFIRINOX or FOLFIRI, l-asparagine, l-leucine, *N*-acetylglutamic acid, l-tyrosine, cellobiose, and l-arginine provided an AUC of 0.972, 0.944, 0.917, 0.889, 0.889, and 0.861, respectively, in responders (SD + PR) vs. non-responders (PD) (Figure 3). In patients treated with gemcitabine + abraxane, inosine, salicylic acid, taurocholic acid, 2-methyl-1-butanol, o-hydroxy hippuric acid, and xanthine provided an AUC of 0.865, 0.838, 0.827, 0.773, 0.769, and 0.758, respectively, in responders (SD + PR) vs. non-responders (PD) (Figure 4).

## 4. Discussion

Oncogenic driver mutations, i.e., *KRAS*, have been shown to reprogram multiple metabolic pathways in pancreatic ductal adenocarcinoma (PDAC) to support growth and proliferation [34]. Unsurprisingly, targeting the metabolic dependences of PDAC has been a focus for the development of novel therapeutic approaches in an otherwise lethal disease [34]. In this post hoc analysis of the prospective PanCax-1 trial (NCT02400398), we evaluated the feasibility of measuring plasma metabolites in a cohort of cachectic subjects with unresectable LAPC or metastatic PC receiving enteral nutrition and cytotoxic chemotherapy. We were successfully able to conduct plasma metabolomics in this population and evaluate the relationship between metabolites involved in multiple metabolic pathways, including glycolysis, glutaminolysis, the tricarboxylic acid (TCA) cycle, fatty acid synthesis, and nucleoside/nucleotide synthesis to responders (PR/SD) and non-responders (PD) to chemotherapy.

In pretreatment (baseline) blood samples, we firstly observed increased levels of taurocholic acid and nucleosides (inosine, inosine 5-triphosphate, and deoxyguanosine 5-triphosphate) in non-responders, while decreased levels of non-essential amino acids (NEAAs; l-asparagine and the oxidized derivative of l-cysteine, l-cystine), uric acid, and phenylpyruvic acid (byproduct of phenylalanine metabolism, an essential amino acid) were associated with PD (Figure 2). Taurocholic acid, a bile acid, has been shown to be elevated nearly 300-fold compared to controls in patients with a biliary obstruction [35]. Biliary obstructions are common in pancreatic cancer subjects, with preclinical evidence to support the idea that bile acids accelerate pancreatic carcinogenesis [36]. Nucleotide synthesis is crucial for supporting PDAC growth and is among the key metabolic pathways altered by *KRAS* mutations [37]. Our results, suggestive of increased taurocholic acid and nucleoside levels at the pretreatment baseline as a poor prognostic indicator of a patient’s response to chemotherapy, are, therefore, consistent with these data. We also observed increased levels of taurocholic acid and the nucleosides xanthine and hypoxanthine in non-responders to gemcitabine-based chemotherapy as well (Figure 2C).

The reprogramming of amino acid metabolism has also been shown to be a hallmark of PDAC progression [38]. Interestingly, we showed that decreases in essential amino acids (EAAs; l-leucine, l-methionine, and l-tryptophan) and NEAAs (l-arginine, l-asparagine, l-serine, and l-tyrosine) in subjects treated with FOLFIRINOX or FOLFIRI was associated with a response of PD to chemotherapy (Figure 2B). We observed a similar association to PD in those with decreased pretreatment levels of NEAAs (Figure 2A). This is opposite to recent efforts to deprive amino acids as a therapeutic strategy, and to studies showing that higher levels of amino acids promote pancreatic tumor growth [38,39]. It is possible that in our cohort of cachectic subjects with APC, lower amino acid levels may be reflective of the hypercatabolic state associated with cancer cachexia [40], which may have compounded the evaluation of amino acid levels as a predictor of chemotherapy response.

We observed that decreased levels of the intermediates of glycolysis (2-phosphoglyceric acid), glutamine metabolism (*N*-acetylglutamic acid), and nucleic acid biosynthesis (2-deoxyguanosine 5-monophosphate, adenosine 5-diphosphate, and *N*-carbamoyl-D l-aspartic acid) were associated with PD after 5-FU-based chemotherapy (Figure 2B). In contrast, increased levels of the intermediates of glycolysis (pyruvate and lactic acid), the TCA cycle (l-glutamate), nucleoside synthesis (xanthine and hypoxanthine), and bile acid metabolism (taurocholic acid) were associated with PD, while decreases in the EAA l-isoleucine was associated with PR/SD after gemcitabine-based chemotherapy (Figure 2C). The findings seen in the subjects treated with gemcitabine-based chemotherapy were consistent with recent literature demonstrating that metabolic reprogramming to support glucose and glutamine metabolism signaling, as well as nucleotide and amino acid synthesis, promotes PDAC survival and progression [37,38,41,42].

Our study is limited due to understanding that plasma metabolites are rather dynamic and can fluctuate due to multiple factors. Additionally, these patients met clinical definitions of cancer cachexia as required for enrollment into the PanCax-1 trial [11]. It is likely that cachexia and chemotherapy treatment impacted the metabolism in these subjects, and this needs to be considered when interpreting our findings [43,44]. We are, however, among the first groups to evaluate metabolomic profiles in cachectic patients with APC who received enteral feeding over 12 weeks. Thus, we have a uniquely homogenous cohort in that all subjects were enterally fed as their primary source of nutrition, which is ideal for evaluating metabolomic signatures that can be influenced by diet. To this end, our study is hypothesis-generating in that we have identified multiple metabolic markers in the pretreatment, 5-FU-based, and gemcitabine-based chemotherapy settings that are potentially predictive of the outcome of chemotherapy. Several of these unique to the FOLFIRINOX/FOLFIRI or gemcitabine + abraxane treatments demonstrated a fairly robust predictive performance based on the multivariate ROC analyses (Figure 3 and Figure 4) and warrant further study. Our findings are also informative for a more focused evaluation into specific metabolic pathways identified as mechanisms for further biomarkers and potential therapeutic development.

For example, the role of essential and non-essential amino acids in promoting PDAC progression and as a mechanism to synergize with standard chemotherapy is being actively explored. One ongoing trial is exploring the utility of amino acid restriction with chemotherapy to evaluate its feasibility and identify metabolic biomarkers to predict the therapeutic response (NCT05078775). As such, amino acids in PDAC subjects remain an attractive group of metabolic markers that should be focused on in future studies of plasma metabolomics. To help focus future areas of investigation, the purpose of Figure 3 and Figure 4 was to filter among the top-ranked metabolites in our cohort that were most predictive of the therapeutic response to specific chemotherapy regimens. As a result, we hope to have identified potential novel and undescribed metabolites and related pathways that may be further explored in preclinical or clinical settings for therapeutic and biomarker development. Other bioenergetic pathway metabolites, including those from glycolysis, glutamine, and nucleic acid metabolism, involve pathways that have long been associated with PDAC carcinogenesis and PDAC progression (as described earlier) and thus warrant a dedicated investigation into their role as biomarkers or as therapeutic targets in PDAC as well.

## 5. Conclusions

The PanCax-1 trial was a prospective, single-institution, single-arm study enrolling cachectic patients with APC to receive 12 weeks of enteral feeding and chemotherapy. In preplanned exploratory analyses, blood samples were collected for plasma metabolite-based biomarker evaluation. We are the first to demonstrate the feasibility of plasma metabolomics in a prospective cohort of APC patients on enteral feeding as their primary source of nutrition. Metabolic signatures unique to 5-FU-based chemotherapy (FOLFIRINOX or FOLFIRI) and gemcitabine + nab-paclitaxel may be predictive of a patient’s response and warrant further study. Several individual metabolic markers demonstrated a predictive performance for a patient’s response (or lack of) to chemotherapy in multivariate ROC analyses. We also uncovered several novel metabolites indicative of bile acid metabolism, nucleic acid biosynthesis, and glycolytic and glutamine metabolism pathways that are worthy of a focused investigation for further biomarkers and potential therapeutic development.

## Figures and Tables

**Figure 1 cancers-15-03020-f001:**
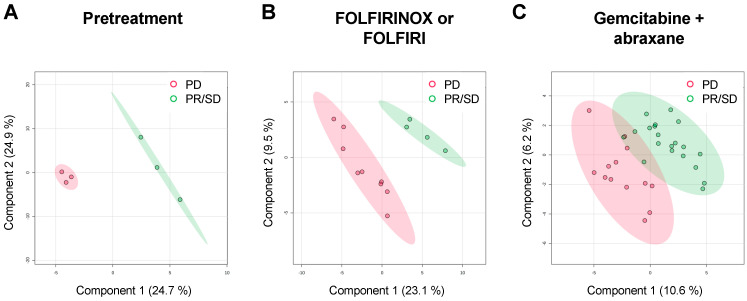
Partial least-squares discriminant analysis (PLS-DA) score plots of metabolomics data from the pretreatment patients (**A**), patients treated with FOLFIRINOX or FOLFIRI (**B**), and patients treated with gemcitabine + abraxane (**C**). Samples from responders (SD + PR) are represented by green circles and samples from non-responders (PD) are depicted as red circles.

**Figure 2 cancers-15-03020-f002:**
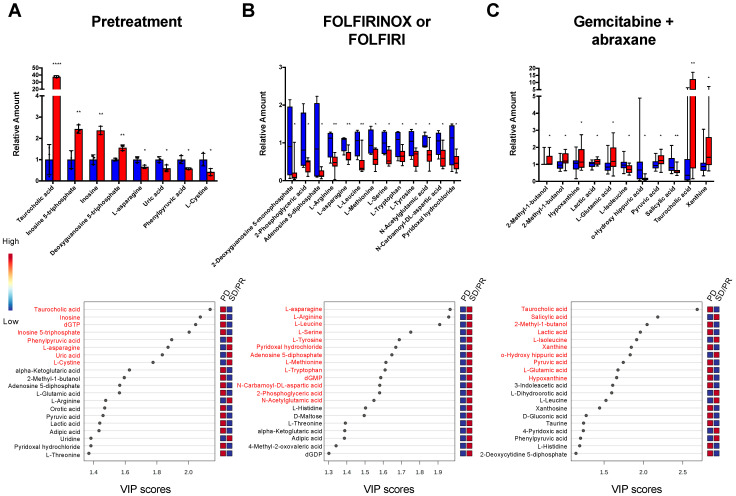
Identification of potential metabolic biomarkers. Upper: metabolites with significantly different levels between responders (SD + PR, blue) and non-responders (PD, red). *Y*-axis is the ratio of metabolite concentrations from PD to SD + PR. Values are means ± SEM, * *p* < 0.05, ** *p* < 0.01 and **** *p* < 0.0001. Lower: top 20 VIP scores and heatmap from PLS-DA of metabolites for responders (SD + PR) vs. non-responders (PD). Red and blue in the heatmap (right) indicate increased and decreased levels, respectively. The name of metabolites that significantly changed in the upper panel are highlighted with red. (**A**) Pretreatment patients (SD + PR, *n* = 3; PD, *n* = 3), (**B**) patients treated with FOLFIRINOX or FOLFIRI (SD + PR, *n* = 4; PD, *n* = 9), and (**C**) patients treated with gemcitabine + abraxane (SD + PR, *n* = 20; PD, *n* = 13).

**Figure 3 cancers-15-03020-f003:**
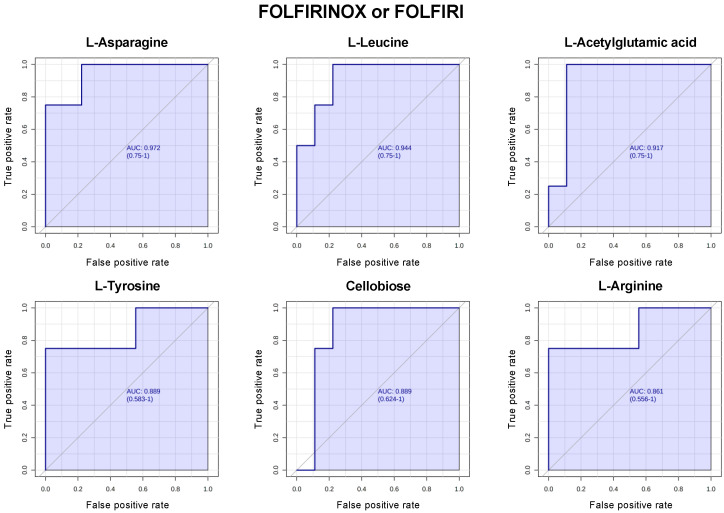
Receiver-operating characteristic (ROC) curve for responders (SD + PR) vs. non-responders (PD) in patients treated with FOLFIRINOX or FOLFIRI. l-asparagine, l-leucine, *N*-acetylglutamic acid, l-tyrosine, cellobiose, and l-arginine provided an AUC of 0.972, 0.944, 0.917, 0.889, 0.889, and 0.861, respectively. The higher the AUC, the better the model is at distinguishing between responders and non-responders.

**Figure 4 cancers-15-03020-f004:**
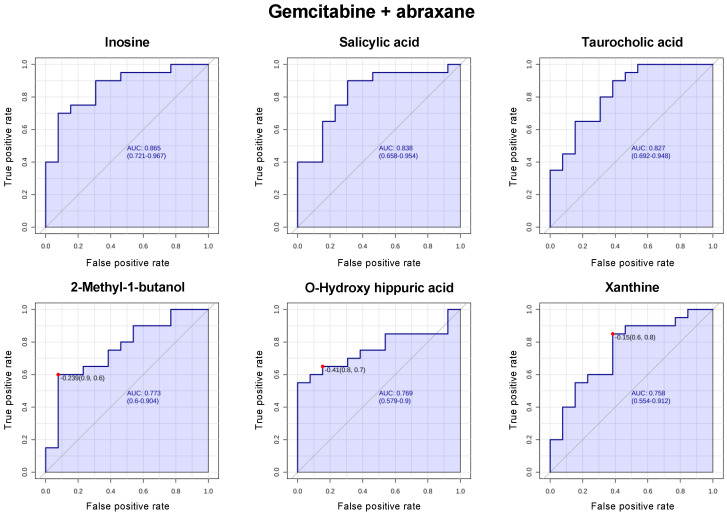
ROC curve for responders (SD + PR) vs. non-responders (PD) in patients treated with gemcitabine + abraxane. Inosine, salicylic acid, taurocholic acid, 2-methyl-1-butanol, o-hydroxy hippuric acid, and xanthine provided an AUC of 0.865, 0.838, 0.827, 0.773, 0.769, and 0.758, respectively.

**Table 1 cancers-15-03020-t001:** Patients and clinical characteristics.

Characteristic	*N* (%) or Mean
Total patients	31
Age	67.1 years old
Gender	
Male	12 (38.7)
Female	19 (61.3)
Race/ethnicity *	
Non-Hispanic White	19 (61.3)
African American	2 (6.4)
Asian/Pacific Islander	7 (22.6)
Hispanic/Latino	7 (22.6)
Other	2 (6.4)
Not reported	1 (3.2)
Line of treatment	
First	22 (71.0)
Second	8 (25.8)
Third	1 (3.2)
CTX regimen	
None	4 (12.9)
Gemcitabine-based	19 (61.3)
Gemcitabine + abraxane	17 (54.8)
Gemcitabine + abraxane + peg-hyaluronidase	1 (3.2)
Gemcitabine	1 (3.2)
5-fluorouracil-based	8 (25.8)
FOLFIRINOX	2 (6.4)
FOLFRI	4 (12.9)
5-FU + Oniyvde + anti-Ilα	2 (6.4)
Best response to CTX	
N/A	4 (12.9)
PR	2 (6.4)
SD	10 (35.3)
PD	15 (48.4)
Median OS	6.53 months
None	4.75 months
Gemcitabine-based	8.83 months
5-fluorouracil-based	5.48 months

* Each subject may identify with more than one ethnicity.

## Data Availability

Not applicable.

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
