# Peer review of "Plasma Metabolomics Predicts Chemotherapy Response in Advanced Pancreatic Cancer"

_cancers, 2023, doi:10.3390/cancers15113020_

Round 1

Reviewer 1 Report

Attempting to evolve metabolic biomarkers in response toward chemo treatments, authors applied ROC curve multivariate analysis of plasma metabolites using LC/MS for responders (SD + PR) vs non-responders (PD) following chemotherapeutic treatments.   Authors recruited 31 cachectic subjects who homogeneously received jejunal tube peptide-based diet for 12 weeks and were planned for palliative chemotherapy.    Despite well planned experimental approaches with convincing data, the following points shall be brought to authors’ attention:

1)    These 31 patients are later categorized to partial response (PR)/stable disease (SD), and progressive disease (PD), according to their chemotherapy response.   Yet, authors shall have denoted the criteria for categorization.

2)    Was the plasma isolated in 0, 6 weeks, and 12 weeks?  Yet, the data seems to present only the end point 12-week.

3)    Authors shall speculate how to apply data to clinical practice.  Are these biomarkers serving as a consequences of chemo or can authors filter the one best metabolites to correspond favorable chemo efficacy?

4)    In order to ignite chemo agent-specific biomarkers, various metabolites displayed in the panels A, B and C of Figure 2, shall be kept in the same order, to view the trend of metabolic fluctuates throughout various chemo treatments.   It is acknowledged that VIP score was originally used for sorting the metabolites. 

5)    What do blue (vs Red) bars indicate in all panels of Figure 3?

6)    Clinical studies beyond the initial 31 cachectic patients shall be expanded, in order to assess if metabolic biomarkers remain valid.

7)    The mechanism explaining the differential metabolic changes shall be speculated and perhaps articulate as a CONCLUSIVE diagram presenting the distinct pathways between various chemo agents and between different responders (SD + PR) vs. non-responders (PD).

8)    Majorities of citations (bibliographies) are old (before year 2020)

9)    Name of the last author on cover page is missing

Reviewer 2 Report

Dear Authors, congratulations for the study. the originality of the study is high, the ideea of investigating a panel of  plasma metabolite instead of a singular is of more utility and offering e better view for orientating further research in the field. I find your article well done and very interesting and progressivist  in this field.

Reviewer 3 Report

Muranaka et al. work provides interesting insights on the hot topic regarding the use of mass spectrometric analyses of patients’ plasma to identify possible biormarker signatures that can predict chemotherapy outcome in PDAC patients. The topic is original and timely, and the experimental setting appears to be correctly framed to support the authors’ objective. Chosen keywords correctly represent the authors’ topic. The introduction part is correctly focused and does not lack essential and important information required for understanding the authors’ work. Methods section is clear and well described. The strength of the study is that is the first of the kind but there are many limitations mentioned by the authors in the discussion. However, to ensure publication suitability, this reviewer raises these major and minor issues as detailed in the checklist below:

Major:

·        Could the authors explain what led to the exclusion of 5 patients (36 patients enrolled, 31 patients actually participating) in the prospective trial PanCax-1?

·        Literature data indicate that pancreatic cancer is more common among men than women and that its incidence is higher in African Americans than in any other ethnic group (in the United States). However, in the present study, 61.3% of participants were female individuals and only 2/31 were African American. Regarding the study design, what are the authors comments regarding the representativeness of their cohort?

·        Could the authors provide Kaplan-Meier plots of the cohort to increase comprehension?

·        PCA score plots obtained from the metabolomics data, despite revealing poor separation as stated in the text, should be added to the Supplementary Information.

·        An additional table for plasma metabolite changes results reporting Univariate test FDR, Multivariate test VIP score and Fold change for each increased/decreased considered metabolite must be added to ensure a better data display. Moreover, a volcano plot showing fold change and FDR should also be added along with their original and normalized concentrations reported as box plots (already present as in figure 3). Finally, a heatmap showing the concentration of the metabolites with the set VIP score, FDR, false discovery rate for each PR/SD vs PD should be also inserted.

·         In figure 2, a clear distinction between significant and non-significant biomarkers must be made (e.g., highlighted). Figure 3 lacks a key of colours used and in its figure legend statistical analyses are missing, affecting results comprehension.

·        Non-significant metabolites analysed but not reported (e.g. p > 0.05, levels below detection threshold, etc) must be added to Supplementary Materials as an .xls file.

·        Results, Discussion and Conclusions do not fully explain Figure 4 and 5 messages, leaving to the reader the explanation to the AUC values obtained from the analysis.

·        In the conclusion, the authors based on the results of the study, should more clearly set further goals and perspectives.

Minor:

·        The manuscript requires a deep check for different kinds of typos, e.g., misplaced or missing parentheses (line 51 and others), acronyms not made explicit (dMRM in line 121, PDAC in line 175, and others) or repeated (PLS-DA in line 177), unmodified (X circles in lines 186-187) or misleading (green instead of blue in line 217)

·        Table 1 inserted in the text is the same as the one in supplementary materials. Do the authors intended to upload a different table? Could the authors insert the better resolution one in text?

·        Figures must be in the same order in which they appear in the text (Figure 3A is cited before Figure 2B). This reviewer suggests to merge figures 2 and 3 and correct the order of appearance.

Quality of English language is fine, only minor spell checks are required.

Round 2

Reviewer 3 Report

The authors successfully addressed my concerns. I am pleased to recommend this work for publication.

As a side note, I would personally add Supplementary Figures 1, 3 and 4 in the main text due to their high visual impact and data display, but I leave the decision to the authors.